# How to promote knowledge transfer within R&D team? An evolutionary game based on prospect theory

**Xiaoya Zhu, Xiaohua Meng** *, **Yanjing Zhang**

School of Politics and Public Administration, Soochow University, Suzhou, China

* tgzy7272@163.com

## Abstract

Knowledge transfer is the basis for R&D teams and enterprises to improve innovation performance, win market competition and seek sustainable development. In order to explore the path to promote knowledge transfer within the R&D team, this study considers the bounded rationality and risk preference of individuals, incorporates prospect theory into evolutionary game, constructs a perceived benefits matrix distinct from the traditional benefits matrix, and simulates the evolutionary game process. The results show that, R&D personnel's knowledge transfer decisions depend on the net income difference among strategies; only if perceived cost is less than the sum of perceived synergy benefit, perceived organization reward value, and perceived organization punishment value, can knowledge be fully shared and transferred within the R&D team. Moreover, R&D personnel's knowledge transfer decisions are interfered by the irrational psychological factors, including overconfidence, reflection, loss avoidance, and obsession with small probability events. The findings help R&D teams achieve breakthroughs in improving the efficiency of knowledge transfer, thereby enhancing the capacity of enterprises for collaborative innovation.

## Introduction

China's strategic document "Made in China 2025" pointed out that the international industrial structure is going through recombination, while China's economic development is being reformed. It is a historical intersection. Thus, Chinese enterprises must seize this opportunity and apply the basic principle "innovation drives development" to change the situation "big but not strong". China's Fourteenth Five-Year Plan (2021–2025) also emphasizes that more attention should be paid to collaborative thinking, development, and innovation.

Knowledge ("Knowledge" in this article includes explicit knowledge and tacit knowledge. Compared with the explicit knowledge, tacit knowledge is difficult to transfer, but trust and close social relationships can effectively promote the transfer of tacit knowledge) transfer is an important means to achieve collaborative innovation [1]. Especially, knowledge transfer within R&D team, as an important organization for enterprises to carry out scientific research and knowledge innovation, is the key way to realize knowledge collaboration and resource complementarity among team members [2–5]. Whether knowledge transfer can be carried out

**Data Availability Statement:** All relevant data are within the manuscript.

**Funding:** This study was supported by the Jiangsu Province Social Science Foundation for Youths (No.21GLC012) and the National Natural Science

Foundation of China (No.72174136).
Conceptualization: Xiaoya Zhu. Data curation:
Xiaohua Meng. Formal analysis: Xiaoya Zhu,
Xiaohua Meng. Preparation of the manuscript:
Xiaoya Zhu, Xiaohua Meng. The funders had no
role in study design, data collection and analysis,
decision to publish, or preparation of the
manuscript.

**Competing interests:** The authors have declared
that no competing interests exist.

continuously and stably determines the team's knowledge innovation ability and the enterprise's scientific research and technological innovation potential, which is of great significance to the rapid development of enterprises and the formation of innovation system [6]. However, in practice, knowledge transfer activities in Chinese enterprises R&D teams are inefficient, R&D personnel's willingness to transfer knowledge and absorptive capacity are insufficient, thus constraining the knowledge transfer process and team collaborative innovation [7]. Accordingly, this situation poses the following questions. how to promote knowledge transfer within R&D teams? What mechanism underlies knowledge transfer? What are the key factors influencing R&D personnel's knowledge transfer decisions?

Scholars have found that subject's characteristics, object's characteristics, and the organizational environment significantly affect knowledge transfer performance. Szulanski (1996), Aladwani (2002), Xu (2014), Huang (2017), and Liu (2018) proposed that the knowledge provider's transfer motivation, knowledge expression ability, knowledge transfer willingness and knowledge teaching ability have significant effects on knowledge transfer performance [8–11]. Cummings (2003), Cao (2012), Ye (2014), Huang (2017), Wang (2017), and Xu (2018) observed that the complexity, embeddedness, concealment, and compatibility of knowledge significantly affect the knowledge transfer effect [11–16]. Awang (2013), Paulsen (2014), Håkonsson (2016), Huang (2017), Ye (2014), Huang (2015), and Liu (2018) posited that knowledge absorptive capacity is the prerequisite for knowledge recipients to use external knowledge, and the key for organizations and enterprises to innovate and maintain competitiveness [11, 14, 17–21].

Furthermore, from the perspective of the "dual" scenario of knowledge transfer, Galbraith (1990), Ramanadhan (2009), Rutten (2016), Zhou (2015), and Liu (2018) pointed out that the geographical distance, psychological distance, and knowledge distance among the participants (individuals or enterprises) play an important role in knowledge transfer performance, and their effects on explicit knowledge and tacit knowledge are quite different [21–25]. From the perspective of "network" scenarios, Greenaway (2015), Zhou (2008), Xu (2018), and Xu (2017) argued that the structure of the social network formed by individuals(e.g., strong connection, weak connection, network centrality, network structure holes, and network density) play a vital role on knowledge transfer performance [16, 26–28].

In addition, Yin (2019, 2022) and Khan (2023) noted that the knowledge spiral has a significant positive effect on new green product development speed and quality, and digital green knowledge creation significantly affects digital green innovation performance [29–31]. This finding provides a perfect theory to understand knowledge transfer better. However, through numerous literature reviews, it is clear that few scholars have studied the deep process mechanism of knowledge transfer within the R&D team the exact conditions for R&D personnel to choose knowledge transfer behaviors. It is a key research gap. According to knowledge management theory, knowledge transfer is a game process and the relationship between subjects is a kind of coopetition [32, 33]. The transfer subjects expect to obtain the return of knowledge transfer while worrying that the existing competitive advantage would be weakened [32–35]. This fact implies that, from the perspective of individuals, knowledge transfer is a risky behavior and can produce a win-win situation while also bring risks [34–37]. As game theory effectively analyzes the decision-making process [38–41], some scholars have begun adopting game theory to explore knowledge transfer behavior [32, 33, 42–45]. However, the traditional evolutionary game theory cannot fully explain a subject's irrationality and risk preference.

According to prospect theory, individuals' risk appetites are inconsistent in the face of gains and losses. They exhibit a significant amount of risk-seeking when facing "losses", but they become risk-averse when facing "gains", which, in turn, influences individual decision-making. Prospect theory can describe and predict behaviors that are inconsistent with traditional

expectation and expected utility theories when people make risky decisions [46, 47]. Zhou (2012) integrated prospect theory into the evolutionary game theory to study the behavior of construction safety management and identified four conditions to eliminate risk-taking behaviors [48]. Shen (2021), based on prospect theory and evolutionary game theory, established an evolutionary game model between local governments and polluting enterprises in the Taihu Lake Basin, and examined the influencing factors of watershed ecological compensation [49].

Accordingly, to conform to the bounded rationality assumption, this study incorporates prospect theory into the analysis process of the evolutionary game and constructs a perceived benefits matrix from a micro perspective. Based on the perceived benefits matrix, we analyze the evolutionary game mechanism of knowledge transfer behavior, explore the boundary conditions for R&D personnel to choose knowledge transfer strategies, and determine the measures to maximize R&D team collaboration benefits.

The contributions of this paper are as follows. From a micro perspective, it reveals the process mechanism of knowledge transfer within an R&D team by integrating prospect theory and evolutionary game theory, clarifies the difference between the perceived and theoretical benefits of the knowledge transfer behavior of R&D personnel, discovers the critical conditions and key factors for the knowledge transfer system to achieve an ideal equilibrium, and proposes the crucial role of R&D personnel's psychological awareness in the knowledge transfer game. Ultimately, this study can help enterprises understand why promoting knowledge transfer in practice is difficult and what measures can effectively motivate R&D personnel to make knowledge transfer decisions.

## An evolutionary game analysis of knowledge transfer

### Evolutionary game model

**Game model construction.** The following related assumptions and parameters are devised to establish an evolutionary game model of knowledge transfer within R&D team [50, 51].

(1) Game player

Regard the R&D team as a system. The system comprises two groups (1 and 2), and the players come from different groups: members 1 and 2, both of which are bounded rationally.

(2) Behavioral strategies of players

Each member's strategies set is discrete in a bounded rational game, with only two choices: knowledge transfer and knowledge non-transfer. Team members adjust to better strategies through continuous imitation and learning. On the one hand, if the perceived benefit of the knowledge transfer strategy is higher, they will stop the knowledge non-transfer actions and choose the knowledge transfer strategy in the next round of the game. On the other hand, if the perceived benefit of the knowledge non-transfer strategy is higher, more members will choose the knowledge non-transfer strategy.

(3) Model hypothesis

**Hypothesis 1:** The choice of strategy depends on perceived gains and losses rather than the actual gains and losses [52]. The characteristics of perceived values are in line with prospect theory. Therefore, this study refers to R&D personnel's psychological perception of gains and losses as prospect value $V = \sum_i \pi(p_i)v(\Delta\omega_i)$, which consists of two parts, a value function $v(x)$ and a weight function $\pi(p)$ [47].

$p_i$ is the probability of event $i$ and the decision weight $\pi(p_i)$ is a function of probability $p_i$. It has the following characteristics: $\pi(0) = 0$, $\pi(1) = 1$. When the value of $p_i$ is small, $\pi(p_i) > p_i$; when the value of $p_i$ is large, $\pi(p_i) < p_i$; and for all $0 < p_1, p_2, r \leq 1$, $\pi(p_1 p_2)/\pi(p_1) \geq \pi(p_1 p_2 r)/\pi(p_1 r)$.

$\Delta\omega_i$ is the difference between the participant's actual benefit and the reference point (The reference point is the reference used by the decision-maker to judge the gains and losses. It can be the current level of the decision-maker's benefit or the level of the decision-maker's expected benefit) when event $i$ occurs. $\Delta\omega_i = \omega_i - \omega_0$. $V(\Delta\omega_i)$ is the value function of $\Delta\omega_i$. It has the following characteristics. When $\Delta\omega_i > 0$, $v''(\Delta\omega_i) < 0$; when $\Delta\omega_i < 0$, $v''(\Delta\omega_i) > 0$, at this time, the curve of the value function $v(x)$ is steeper.

The formulas of the value function $v(x)$ and weight function $\pi(p)$ are as follows:

$$v(x) = \begin{cases} x^\alpha & , x \geq 0 \\ -\lambda(-x)\beta & , x < 0 \end{cases} \tag{1}$$

$$\pi(p) = \frac{p^\gamma}{[p^\gamma + (1-p)^\gamma]^{1/\gamma}} \tag{2}$$

where $x$ represents the value of the selected solution. $\alpha$, $B(0 < \alpha, \beta < 1)$ represents the risk preference coefficient. The larger the $\alpha$, $\beta$, the more adventurous the decision-maker is. $\lambda(\lambda \geq 1)$ represents the loss avoidance coefficient. The larger the $\lambda$, the more sensitive the decision-maker is to losses.

**Hypothesis 2:** Owing to the limitations of the knowledge teaching and absorptive capacities of R&D personnel, the amount of knowledge that the knowledge provider decides to transfer, expressed as $k_i$, is linearly correlated with the amount of knowledge that the knowledge receiver can acquire, called direct benefit, and expressed as $U_j^1$. The correlation coefficient can be called transfer coefficient, expressed as $a_j(0 < a_j < 1)$. If Member 2 chooses knowledge transfer action, the direct benefit of Member 1 can be expressed as $U_1^1 = a_1 k_2$.

**Hypothesis 3:** Knowledge has aggregation effects. When a player decides to transfer their knowledge to another player, new knowledge is created by the receiver through friction between the receiver's own knowledge and foreign knowledge. This study uses $U_j^2$ to represent the knowledge aggregation benefit, $s_j$ to represent the knowledge stock, and $b_j$ to represent the aggregation coefficient, which mainly depends on knowledge learning and application abilities. If Member 2 chooses the knowledge transfer action, the aggregation benefit for Member 1 can be expressed as $U_1^2 = b_1 s_1 k_2$.

**Hypothesis 4:** Knowledge has synergy effects. When both game players actively participate in knowledge transfer, the exchange and collision of knowledge between the two participants will create new knowledge. This study uses $U_j^3$ to represent the knowledge synergy benefit; $d_j$ to represent the synergy coefficient, which mainly depends on the knowledge complementarity, knowledge creation and cooperation abilities of team members; $m$ and $n$ to represent the elastic coefficient of the amount of knowledge transfer, and $m, n > 0$, $m + n = 1$. Then, if members 1 and 2 choose knowledge transfer actions, the synergy benefit for Member 1 can be expressed as $U_1^3 = d_1 k_1^m k_2^n$.

**Hypothesis 5:** Knowledge transfer costs. The player that chooses the knowledge transfer strategy will pay material and spiritual costs, such as time, energy, opportunity, and competitive advantage. This study uses $C_j$ to represent the cost of knowledge transfer, and $c_j$ to represent the cost coefficient of $j$. Then the cost of Member 1 can be expressed as $C_1 = c_1 k_1$.

**Hypothesis 6:** In knowledge transfer activities, organizations often intervene in R&D personnel's actions. This study incorporates an organizational system into the game process, including organizational incentives and punishments systems. In this study, $ek_j$ is used to represent the reward value, including explicit incentives (such as job promotion, material rewards, financial rewards, and honorary recognition) and implicit incentives (including reputation and potential opportunities), where $e$ represents the organization's reward coefficient for knowledge transfer action; $\varphi$ represents the punishments imposed by the organization on opportunistic, free-riding and other knowledge non-transfer actions, including mental punishment and economic punishment. $\eta$ represents the probability that an organization discovers and punishes a knowledge non-transfer action.

**Perceived benefits matrix.**   (1) When the strategies of game players (members 1 and 2) are knowledge transfer and knowledge transfer, according to **Hypothesis** 2–6, it can be deduced that the direct benefit of Member 1 is $U_1^1 = a_1 k_2$, the aggregation benefit is $U_1^2 = b_1 s_1 k_2$, the synergy benefit is $U_1^3 = d_1 k_1^m k_2^n$, the cost is $C_1 = c_1 k_1$, and the organizational reward value is $ek_1$. At this time, the probability of the event that "Member 1 chooses knowledge transfer" is $p_1 = 1$, the probability of the event that "Member 2 chooses knowledge transfer" is $p_2 = 1$, and the probability of the event that "both players choose knowledge transfer" is $p_3 = 1$.

Then, according to **Hypothesis** 1, Member 1's perceived direct benefit is

$$V_1^1 = \pi(p_2)v(U_1^1) + \pi(1 - p_2)v(0) = \pi(1)v(a_1 k_2) + \pi(0)v(0) = v(a_1 k_2) \quad (3)$$

Member 1's perceived aggregation benefit is

$$V_1^2 = \pi(p_2)v(U_1^2) + \pi(1 - p_2)v(0) = \pi(1)v(b_1 s_1 k_2) + \pi(0)v(0) = v(b_1 s_1 k_2) \quad (4)$$

Member 1's perceived synergy benefit is

$$V_1^3 = \pi(p_2)v(U_1^3) + \pi(1 - p_2)v(0) = \pi(1)v(d_1 k_1^m k_2^n) + \pi(0)v(0) = v(d_1 k_1^m k_2^n) \quad (5)$$

Member 1's perceived cost is

$$V_1^4 = \pi(p_1)v(C_1) + \pi(1 - p_1)v(0) = \pi(1)v(c_1 k_1) + \pi(0)v(0) = v(c_1 k_1) \quad (6)$$

Member 1's perceived organizational reward value is

$$V_1^5 = \pi(p_1)v(ek_1) + \pi(1 - p_1)v(0) = \pi(1)v(ek_1) + \pi(0)v(0) = v(ek_1) \quad (7)$$

In this case, Member 1's total perceived benefit is

$$V_1 = V_1^1 + V_1^2 + V_1^3 - V_1^4 + V_1^5 = v(a_1 k_2) + v(b_1 s_1 k_2) + v(d_1 k_1^m k_2^n) - v(c_1 k_1) + v(ek_1) \quad (8)$$

Similarly, Member 2's total perceived benefit is

$$V_2 = V_2^1 + V_2^2 + V_2^3 - V_2^4 + V_2^5 = v(a_2 k_1) + v(b_2 s_2 k_1) + v(d_2 k_1^m k_2^n) - v(c_2 k_2) + v(ek_2) \quad (9)$$

(2) When the strategies of players in the game are knowledge transfer and knowledge non-transfer, according to **Hypothesis** 3–6, it can be deduced that the direct benefit, aggregation benefit and synergy benefit of Member 1 are all 0. The cost of choosing the knowledge transfer strategy is $C_1 = c_1 k_1$, and the organization will give a reward $ek_1$. At this time, the probability of the event that "Member 1 chooses knowledge transfer" is $p_1 = 1$, the probability of the event that "Member 2 chooses knowledge transfer" is $p_2 = 0$, and the probability of the event that "both players choose knowledge transfer" is $p_3 = 0$.

Thus, according to **Hypothesis** 1, Member 1's perceived cost is

$$V_1^4 = \pi(p_1)v(C_1) + \pi(1 - p_1)v(0) = \pi(1)v(c_1k_1) + \pi(0)v(0) = v(c_1k_1) \tag{10}$$

Member 1's perceived organizational reward value is

$$V_1^5 = \pi(p_1)v(ek_1) + \pi(1 - p_1)v(0) = \pi(1)v(ek_1) + \pi(0)v(0) = v(ek_1) \tag{11}$$

In this case, Member 1's total perceived benefit is

$$V_1 = -V_1^4 + V_1^5 = -v(c_1k_1) + v(ek_1) \tag{12}$$

Similarly, according to **Hypothesis** 3–6, in this case, the synergy benefit and cost of Member 2 are 0, the direct benefit is $U_2^1 = a_2k_1$, and the aggregation benefit is $U_2^2 = b_2s_2k_1$. Member 2 is punished by the organization with probability $\eta$, and the punishment value is $\varphi$.

Thus, Member 2's perceived direct benefit is

$$V_2^1 = \pi(p_1)v(U_2^1) + \pi(1 - p_1)v(0) = \pi(1)v(a_2k_1) + \pi(0)v(0) = v(a_2k_1) \tag{13}$$

Member 2's perceived aggregation benefit is

$$V_2^2 = \pi(p_1)v(U_2^2) + \pi(1 - p_1)v(0) = \pi(1)v(b_2s_2k_1) + \pi(0)v(0) = v(b_2s_2k_1) \tag{14}$$

Member 2's perceived organizational punishment value is

$$V_2^6 = \pi(\eta)v(\varphi) + \pi(1 - \eta)v(0) = \pi(\eta)v(\varphi) \tag{15}$$

In this case, Member 2's total perceived benefit is

$$V_2 = V_2^1 + V_2^2 - V_2^6 = v(a_2k_1) + v(b_2s_2k_1) - \pi(\eta)v(\varphi) \tag{16}$$

(3) When the strategies of players in the game are knowledge non-transfer and knowledge transfer, according to **Hypothesis** 3–6, it can be seen that the synergy benefit and cost of Member 1 are 0, the direct benefit is $U_1^1 = a_1k_2$, and the aggregation benefit is $U_1^2 = b_1s_1k_2$. Member 1 is punished by the organization with probability $\eta$, and the punishment value is $\varphi$. At this time, the probability of the event that "Member 1 chooses knowledge transfer" is $p_1 = 0$, the probability of the event that "Member 2 chooses knowledge transfer" is $p_2 = 1$, and the probability of the event "both participants choose knowledge transfer" is $p_3 = 0$.

Thus, according to **Hypothesis** 1, Member 1's perceived direct benefit is

$$V_1^1 = \pi(p_2)v(U_1^1) + \pi(1 - p_2)v(0) = \pi(1)v(a_1k_2) + \pi(0)v(0) = v(a_1k_2) \tag{17}$$

Member 1's perceived aggregation benefit is

$$V_1^2 = \pi(p_2)v(U_1^2) + \pi(1 - p_2)v(0) = \pi(1)v(b_1s_1k_2) + \pi(0)v(0) = v(b_1s_1k_2) \tag{18}$$

Member 1's perceived organizational punishment value is

$$V_1^6 = \pi(\eta)v(\varphi) + \pi(1 - \eta)v(0) = \pi(\eta)v(\varphi) \tag{19}$$

In this case, Member 1's total perceived benefit is

$$V_1 = V_1^1 + V_1^2 - V_1^6 = v(a_1k_2) + v(b_1s_{1,0}k_2) - \pi(\eta)v(\varphi) \tag{20}$$

Similarly, according to **Hypothesis** 3–6, it can be seen that the direct benefit, aggregation benefit and synergy benefit of Member 2 are all 0, the cost of choosing the knowledge transfer strategy is $C_2 = c_2k_2$, and the organization will give a reward $ek_2$.

Thus, Member 2's perceived cost is

$$V_2^4 = \pi(p_2)v(C_2) + \pi(1-p_2)v(0) = \pi(1)v(c_2 k_2) + \pi(0)v(0) = v(c_2 k_2) \tag{21}$$

Member 2's perceived organizational reward value is

$$V_2^5 = \pi(p_2)v(ek_2) + \pi(1-p_2)v(0) = \pi(1)v(ek_2) + \pi(0)v(0) = v(ek_2) \tag{22}$$

In this case, Member 1's total perceived benefit is

$$V_2 = -V_2^4 + V_2^5 = -v(c_2 k_2) + v(ek_2) \tag{23}$$

(4) When the strategies of players in the game are knowledge non-transfer and knowledge non-transfer, then according to **Hypothesis** 3–6, neither player has any benefits or cost, but under probability $\eta$, they will be punished by the organization, and the punishment value is $\varphi$.

Thus, according to **Hypothesis** 1, Member 1's total perceived benefit is

$$V_1 = -V_1^6 = -\pi(\eta)v(\varphi) \tag{24}$$

Member 2's total perceived benefit is

$$V_2 = -V_2^6 = -\pi(\eta)v(\varphi) \tag{25}$$

By combining the above **hypothesis** and analyses, we construct the perceived benefits matrix of the knowledge transfer game between the members of the R&D team, as shown in Table 1.

**Game model solution and analysis.** According to Table 1 and evolutionary game theory, which incorporates the idea of replication dynamics [53], we can obtain the expected value of the perceived benefits of each strategy.

In the initial stage of the game, if the probabilities of Member 1 and Member 2 choosing the knowledge transfer strategy are $x$ and $y$ respectively, the expected value of the perceived benefits of Member 1 choosing the knowledge transfer strategy is

$$\begin{aligned} E_1 &= y[v(a_1 k_2) + v(b_1 s_1 k_2) + v(d_1 k_1^m k_2^n) - v(c_1 k_1) + v(ek_1)] + (1-y)[-v(c_1 k_1) + v(ek_1)] \\ &= y[v(a_1 k_2) + v(b_1 s_1 k_2) + v(d_1 k_1^m k_2^n)] - v(c_1 k_1) + v(ek_1) \end{aligned} \tag{26}$$

The expected value of the perceived benefits of Member 1 choosing the knowledge non-transfer strategy is

$$\begin{aligned} E_1' &= y[v(a_1 k_2) + v(b_1 s_1 k_2) - \pi(\eta)v(\varphi)] + (1-y)[-\pi(\eta)v(\varphi)] \\ &= y[v(a_1 k_2) + v(b_1 s_1 k_2)] - \pi(\eta)v(\varphi) \end{aligned} \tag{27}$$

**Table 1. Perceived benefits matrix for knowledge transfer within the R & D team.**

| | | Member 2 | | Member 2 |
| --- | --- | --- | --- | --- |
| | | Transfer | | Non-transfer |
| Member1 | Transfer | $v(a_1 k_2) + v(b_1 s_{1,0} k_2) + v(d_1 k_1^m k_2^n) - v(c_1 k_1) + v(ek_1)$, $v(a_2 k_1) + v(b_2 v_{2,0} k_1) + v(d_2 k_1^m k_2^n) - v(c_2 k_2) + v(ek_2)$ | | $-v(c_1 k_1) + v(ek_1)$, $v(a_2 k_1) + v(b_2 s_{2,0} k_1) - \pi(\eta)v(\varphi)$ |
| | Non-transfer | $v(a_1 k_2) + v(b_1 s_{1,0} k_2) - \pi(\eta)v(\varphi)$, $-v(c_2 k_2) + v(ek_2)$ | | $-\pi(\eta)v(\varphi)$, $-\pi(\eta)v(\varphi)$ |

Therefore, the average value of the perceived benefits of Member 1 is

$$
\begin{aligned}
\overline{E}_1 &= xE_1 + (1-x)E'_1 \\
&= xyv(d_1k_1^m k_2^n) + y[v(a_1k_2) + v(b_1s_1k_2)] + x[\pi(\eta)v(\varphi) - v(c_1k_1) + v(ek_1)] - \pi(\eta)v(\varphi)
\end{aligned}
\tag{28}
$$

Similarly, the expected value of the perceived benefits of Member 2 choosing the knowledge transfer strategy is

$$
\begin{aligned}
E_2 &= x[v(a_2k_1) + v(b_2s_2k_1) + v(d_2k_1^m k_2^n) - v(c_2k_2) + v(ek_2)] + (1-x)[-v(c_2k_2) + v(ek_2)] \\
&= x[v(a_2k_1) + v(b_2s_2k_1) + v(d_2k_1^m k_2^n)] - v(c_2k_2) + v(ek_2)
\end{aligned}
\tag{29}
$$

The expected value of the perceived benefits of Member 2 choosing the knowledge non-transfer strategy is

$$
\begin{aligned}
E'_2 &= x[v(a_2k_1) + v(b_2s_2k_1) - \pi(\eta)v(\varphi)] + (1-x)[-\pi(\eta)v(\varphi)] \\
&= x[v(a_2k_1) + v(b_2s_2k_1)] - \pi(\eta)v(\varphi)
\end{aligned}
\tag{30}
$$

Therefore, the average value of the perceived benefits of Member 2 is

$$
\begin{aligned}
\overline{E}_2 &= yE_2 + (1-y)E'_2 \\
&= xyv(d_2k_1^m k_2^n) + x[v(a_2k_1) + v(b_2s_2k_1)] + y[\pi(\eta)v(\varphi) - v(c_2k_2) + v(ek_2)] - \pi(\eta)v(\varphi)
\end{aligned}
\tag{31}
$$

As team members are rationally bounded and their learning speed is slow, they can only adjust their strategies based on the multiple game results. This dynamic adjustment mechanism is similar to the "replication dynamic" in the process of biological dynamic evolution [54, 55]; that is, if the average value of perceived benefits of a particular strategy is higher than the one of the mixed strategies, more and more members will tend this strategy. According to this replication dynamic evolution method, assuming that the adjustment speed of the ratio of different strategies is positively correlated with the difference between the average value of the perceived benefits of the strategy and that of the mixed strategy, the replication dynamic functions of $x$ and $y$ can be expressed as

$$
\frac{dx}{dt} = x(E_1 - \overline{E}_1) = x(1-x)[y \cdot v(d_1k_1^m k_2^n) - v(c_1k_1) + v(ek_1) + \pi(\eta)v(\varphi)]
\tag{32}
$$

$$
\frac{dy}{dt} = y(E_2 - \overline{E}_2) = y(1-y)[x \cdot v(d_2k_1^m k_2^n) - v(c_2k_2) + v(ek_2) + \pi(\eta)v(\varphi)]
\tag{33}
$$

A point $(x, y)$ on the solution curve of the replication dynamic functions corresponds to a set of mixed strategies in the evolutionary game process, and the equilibrium point of the replication dynamic system corresponds to the equilibrium point of the evolutionary game [56]. If $\frac{dx}{dt} = 0, \frac{dy}{dt} = 0$, we can get:

$$
\begin{aligned}
x_1^* = 0, x_2^* = 1, or\ y^* &= \frac{v(c_1k_1) - v(ek_1) - \pi(\eta)v(\varphi)}{v(d_1k_1^m k_2^n)} \\
y_1^* = 0, y_2^* = 1, or\ x^* &= \frac{v(c_2k_2) - v(ek_2) - \pi(\eta)v(\varphi)}{v(d_2k_1^m k_2^n)}
\end{aligned}
\tag{34}
$$

Then, the partial equilibrium points in the set $R = \{(x, y) | 0 \le x \le 1, 0 \ge y \le 1\}$ are (0,0), (0,1), (1,0), (1,1), $(x^*, y^*)(0 < x^*, y^* < 1)$.

## Evolutionary stability analysis

According to the method proposed by Friedman [53], the evolutionary stability strategy (ESS) of the differential function system can be obtained by analyzing the partial stability of its Jacobian matrix.

Hence, the partial derivative of the differential functions is solved to obtain the Jacobian matrix as follows:

$$J = \begin{bmatrix} (1-2x)[y \cdot v(d_1 k_1^m k_2^n) - v(c_1 k_1) + v(ek_1) + \pi(\eta)v(\varphi)] & x(1-x)v(d_1 k_1^m k_2^n) \\ y(1-y)v(d_2 k_1^m k_2^n) & (1-2y)[x \cdot v(d_2 k_1^m k_2^n) - v(c_2 k_2) + v(ek_2) + \pi(\eta)v(\varphi)] \end{bmatrix} \quad (35)$$

Subsequently, the determinant of $J$ is

$$\begin{aligned} DetJ &= (1-2x)(1-2y)[y \cdot v(d_1 k_1^m k_2^n) - v(c_1 k_1) + v(ek_1) + \pi(\eta)v(\varphi)][x \cdot v(d_2 k_1^m k_2^n) \\ &\quad - v(c_2 k_2) + v(ek_2) + \pi(\eta)v(\varphi)] - xy(1-x)(1-y)v(d_1 k_1^m k_2^n)v(d_2 k_1^m k_2^n) \end{aligned} \quad (36)$$

The trace of $J$ is

$$\begin{aligned} TrJ &= (1-2x)[y \cdot v(d_1 k_1^m k_2^n) - v(c_1 k_1) + v(ek_1) + \pi(\eta)v(\varphi)] \\ &\quad + (1-2y)[x \cdot v(d_2 k_1^m k_2^n) - v(c_2 k_2) + v(ek_2) + \pi(\eta)v(\varphi)] \end{aligned} \quad (37)$$

Within a R&D team, the more members actively participate in knowledge transfer activities, the more conducive they are to the circulation and innovation of team knowledge. Therefore, based on overall interests, it is ideal for all members to choose a knowledge transfer strategy. The results of the partial stability analysis are listed in Table 2. Accordingly, we find that only when the two conditions

$$v(d_1 k_1^m k_2^n) + v(ek_1) + \pi(\eta)v(\varphi) > v(c_1 k_1)$$
$$v(d_2 k_1^m k_2^n) + v(ek_2) + \pi(\eta)v(\varphi) > v(c_2 k_2)$$

are met at the same time, would the system converge to point (1, 1). All members actively participate in knowledge transfer activities, thereby enabling a high degree of collaboration and a continuous and stable flow of knowledge.

As $TrJ|(x^*y^*) = 0$ is always established, the conventional method to analyze partial stability is invalid. This study adopts differential analysis to judge the stability of the point $(x^*y^*)$.

**Table 2. Partial stability analysis results.**

| $(x,y)$ | DetJ | Signs | TrJ | Signs | Results |
|---|---|---|---|---|---|
| (0,0) | $[-v(c_1 k_1) + v(ek_1) + \pi(\eta)v(\varphi)]$ $[-v(c_2 k_2) + v(ek_2) + \pi(\eta)v(\varphi)]$ | uncertainty | $[-v(c_1 k_1) + v(ek_1) + \pi(\eta)v(\varphi)]$ $+[-v(c_2 k_2) + v(ek_2) + \pi(\eta)v(\varphi)]$ | uncertainty | Unstable point |
| (0,1) | $-[v(d_1 k_1^m k_2^n) - v(c_1 k_1) + v(ek_1) + \pi(\eta)v(\varphi)] \cdot$ $[-v(c_2 k_2) + v(ek_2) + \pi(\eta)v(\varphi)]$ | uncertainty | $[v(d_1 k_1^m k_2^n) - v(c_1 k_1) + v(ek_1) + \pi(\eta)v(\varphi)]$ $[-v(c_2 k_2) + v(ek_2) + \pi(\eta)v(\varphi)]$ | uncertainty | Unstable point |
| (1,0) | $-[-v(c_1 k_1) + v(ek_1) + \pi(\eta)v(\varphi)]$ $[v(d_2 k_1^m k_2^n) - v(c_2 k_2) + v(ek_2) + \pi(\eta)v(\varphi)]$ | uncertainty | $-[-v(c_1 k_1) + v(ek_1) + \pi(\eta)v(\varphi)]$ $+[x \cdot v(d_2 k_1^m k_2^n) - v(c_2 k_2) + v(ek_2) + \pi(\eta)v(\varphi)]$ | uncertainty | Unstable point |
| (1,1) | $[v(d_1 k_1^m k_2^n) - v(c_1 k_1) + v(ek_1) + \pi(\eta)v(\varphi)] \cdot$ $[v(d_2 k_1^m k_2^n) - v(c_2 k_2) + v(ek_2) + \pi(\eta)v(\varphi)]$ | + | $-[v(d_1 k_1^m k_2^n) - v(c_1 k_1) + v(ek_1) + \pi(\eta)v(\varphi)]$ $-[v(d_2 k_1^m k_2^n) - v(c_2 k_2) + v(ek_2) + \pi(\eta)v(\varphi)]$ | - | ESS |

Differentiating Functions (32) and (33) with respect to $x$ and $y$ respectively, and bringing the point $x^*y^*$ into them, the results are as follows:

$$\frac{dx}{dt} \bigg/ dy = x(1-x)v(d_1 k_1^m k_2^n)$$
$$= \frac{[v(c_2 k_2) - v(ek_2) - \pi(\eta)v(\varphi)][v(d_2 k_1^m k_2^n) - v(c_2 k_2) + v(ek_2) + \pi(\eta)v(\varphi)]}{v^2(d_2 k_1^m k_2^n)} \cdot v(d_1 k_1^m k_2^n) \tag{38}$$

$$\frac{dy}{dt} \bigg/ dx = y(1-y)v(d_2 k_1^m k_2^n)$$
$$= \frac{[v(c_1 k_1) - v(ek_1) - \pi(\eta)v(\varphi)][v(d_1 k_1^m k_2^n) - v(c_1 k_1) + v(ek_1) + \pi(\eta)v(\varphi)]}{v^2(d_1 k_1^m k_2^n)} \cdot v(d_2 k_1^m k_2^n) \tag{39}$$

As $x, y \in [0,1]$, to make the point

$$(x^*, y^*) = \left( \frac{v(c_2 k_2) - v(ek_2) - \pi(\eta)v(\varphi)}{v(d_2 k_1^m k_2^n)}, \frac{v(c_1 k_1) - v(ek_1) - \pi(\eta)v(\varphi)}{v(d_1 k_1^m k_2^n)} \right) \tag{40}$$

meaningful, there must be

$$0 \leq \frac{v(c_2 k_2) - v(ek_2) - \pi(\eta)v(\varphi)}{v(d_2 k_1^m k_2^n)}, \frac{v(c_1 k_1) - v(ek_1) - \pi(\eta)v(\varphi)}{v(d_1 k_1^m k_2^n)} \leq 1 \tag{41}$$

That is,

$$v(ek_1) + \pi(\eta)v(\varphi) \leq v(c_1 k_1) \leq v(d_1 k_1^m k_2^n) + v(ek_1) + \pi(\eta)v(\varphi)$$
$$v(ek_2) + \pi(\eta)v(\varphi) \leq v(c_2 k_2) \leq v(d_2 k_1^m k_2^n) + v(ek_2) + \pi(\eta)v(\varphi)$$

Therefore, by combining the above two conditions, it can be deduced that Functions (38) and (39) are positive, thus the point $(x^*y^*)$ is an unstable point.

The overall dynamic evolution phase diagram of the system is shown in Fig 1.

## Results discussion

The analysis above reveals that only when

$$v(d_1 k_1^m k_2^n) + v(ek_1) + \pi(\eta)v(\varphi) > v(c_1 k_1),$$
$$v(d_2 k_1^m k_2^n) + v(ek_2) + \pi(\eta)v(\varphi) > v(c_2 k_2)$$

can the system reach the ideal equilibrium state of "high collaboration level among R & D personnel, and continuous and stable flow of knowledge". This result shows that the perceived cost of knowledge transfer between R&D personnel cannot be higher than the sum of the perceived synergy benefit and the perceived organizational reward and punishment. However, because game players are bounded rationally, they cannot utilize the existing information maximally, but rely excessively on intuitive and subjective judgments, which ultimately leads to systematic deviations, including overconfidence, reflection effect, loss avoidance effect and obsession with a small probability event effect. These deviations make it more difficult for the evolutionary system to converge to equilibrium point E (1,1). The specific analysis is as follows.

1. Impact of overconfidence. Overconfidence refers to the fact that individuals tend to overestimate their knowledge stock, abilities or the accuracy of their own information [57]. Owing to overconfidence, R&D personnel tend to have a strong sense of self-knowledge

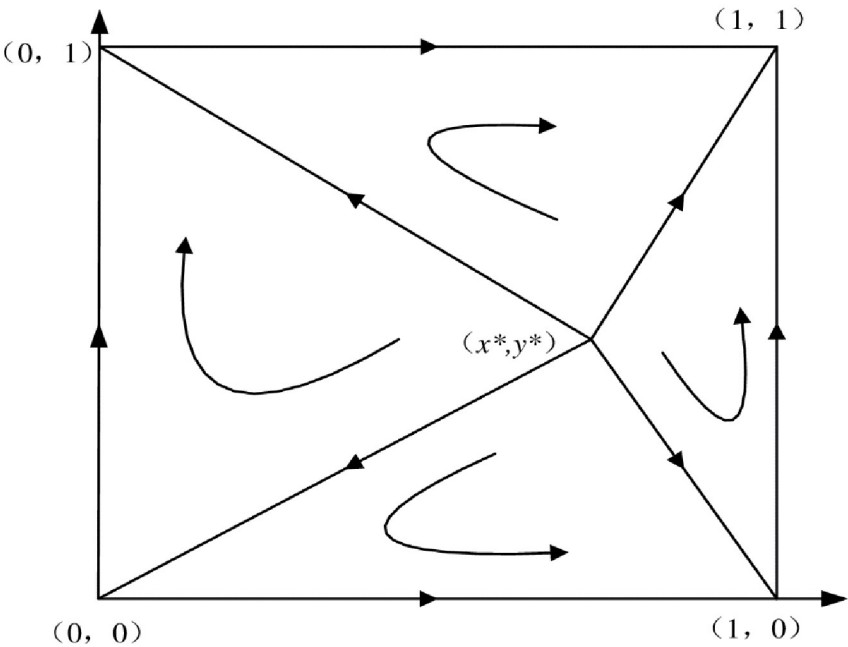

**Fig 1. System evolution phase diagram.**

protection and worry that knowledge transfer will lose their existing competitive advantages. Therefore, both participants in the game tend to amplify the losses caused by knowledge transfer action, have low expectations for the knowledge synergy benefit, and will not even open their hearts completely when exchanging knowledge. Clearly, this scenario is not conducive to satisfy these two conditions.

2. According to the loss avoidance effect, decision-makers are usually more sensitive to losses than gains. When $U < 0$, $v(U)$ is a concave function. Faced with a complex and uncertain environment within the R&D team, R&D personnel usually overestimate the cost of choosing a knowledge transfer strategy, and their perceived cost is higher than the actual cost: $v(c_i k_i) > c_i k_i$. Clearly, this situation is not conducive to satisfy these two conditions.

3. The loss avoidance effect also shows that, when $U > 0$, $v(U)$ is a convex function. In an R&D team, when the personnel judge the value of strategies, they tend to underestimate the benefits of the knowledge transfer strategy, including the knowledge synergy benefit and organizational reward: $v(d_i k_i^m k_j^n) < d_i k_i^m k_j^n$, $v(e k_i) < e k_i$. Evidently, this situation is not conducive to the realization of these two conditions either.

4. According to the reflection effect, the decision-makers' preferences for losses and gains are asymmetrical. When facing the prospect of possible losses, there is a tendency to pursue risk, while when facing the prospect of possible gains, there is a tendency to avoid risk [46]. Furthermore, according to the obsession with a small probability event effect, decision-makers are biased when judging the value of strategies, thinking that they would encounter negative events with a lower probability than others [58, 59]. Therefore, when faced with uncertain organizational punishments, R&D personnel are more willing to take risks, underestimate probability, and pin hope on a fluke. This scenario is also not conducive to the system converging to the equilibrium point E (1,1).

In summary, the combined effects of over confidence, reflection effect, loss avoidance effect, and obsession with a small probability event effect overestimate the cost of knowledge transfer, while underestimate knowledge synergy benefit, organizational reward value, and the probability of organizational punishment. Additionally, both participants of the game prefer to take risks and choose the knowledge non-transfer strategy. They make it difficult to meet the two conditions for continuous knowledge transfer within the enterprise R&D team. In the actual knowledge transfer process within R&D teams, even if the objective conditions for knowledge transfer under the traditional game perspective are met, the bounded rationality and perceived differences in different types of risks may still make the game system unable to converge to the ideal equilibrium.

## Numerical simulation

According to the analysis results of the evolutionary game, within R&D teams, the perceived cost, perceived synergy benefit, perceived organizational reward value, and perceived organizational punishment determine the final evolution result of the knowledge transfer system to a certain extent. This study uses MATLAB to simulate the evolution process of knowledge transfer within R&D teams and reveal the influence paths and modes of the four key factors on knowledge transfer.

The fundamental purpose of knowledge transfer is to realize the full sharing and circulation of knowledge in R&D teams. Hence, this study uses the proportion $y$ ([0,1]) of the number of R&D personnel who adopt knowledge transfer strategies to the total number of R&D personnel in the R&D team as the metrics. In addition, it tests each set of parameters 50 times to obtain stable simulation results.

Based on the actual situation and the basic assumptions of the model, the initial values of simulation parameters were set as $s_j = 10$, $k_j = 1$, $d_j = 0.1$, $c_j = 0.5$, $e = 0.2$, $\varphi = 0.4$, $\eta = 0.5$, $\alpha_j = 0.5$, $b_j = 0.02$.

1. Impact of perceived cost $V_i^4$ of knowledge transfer on evolutionary results. $V_i^4$ was set as 0.2, 0.4, 0.6 and 0.8. As Fig 2 illustrates, there is a critical value between 0.6 and 0.8. When $V_i^4$ is less than the critical value, $y$ converges to 1, but the increase of $V_i^4$ can slow down the speed of $y$ converging to 1; when $V_i^4$ is more than the critical value, $y$ converges to 0. Therefore, reducing the perceived cost of knowledge transfer between R&D personnel can effectively promote knowledge transfer.

2. Impact of perceived knowledge synergy benefit $V_i^3$ on the evolutionary results. $V_i^3$ was set as 0.1, 0.15, 0.2 and 0.25. As Fig 3 illustrates, there is a critical value between 0.2 and 0.25. When $V_i^3$ is less than the critical value, $y$ converges to 0, and when $V_i^3$ is more than the critical value, $y$ converges to 1. Therefore, R&D personnel can be effectively motivated to choose knowledge transfer strategies only when the perceived knowledge synergy benefit increases above a critical value.

3. Impact of the perceived organizational reward value $V_i^5$ on evolutionary results. $V_i^5$ was set as 0.1, 0.2, 0.3 and 0.4. As Fig 4 illustrates, there is a critical value between 0.2 and 0.3. When $V_i^5$ is less than the critical value, $y$ converges to 0, and the increase of $V_i^5$ reduces the speed at which $y$ converges to 0. When $V_i^5$ is more than the critical value, $y$ converges to 1, and the increase of $V_i^5$ can make $y$ converge to 1 faster. Therefore, improving the perceived organizational reward value of R&D team members can prompt more R&D personnel to pay attention to knowledge transfer activities.

4. Impact of perceived organizational punishment value $V_i^6$ on evolutionary results. $V_i^6$ was set as 0.1, 0.2, 0.3 and 0.4. Fig 5 reveals a critical value between 0.1 and 0.2. When $V_i^6$ is less

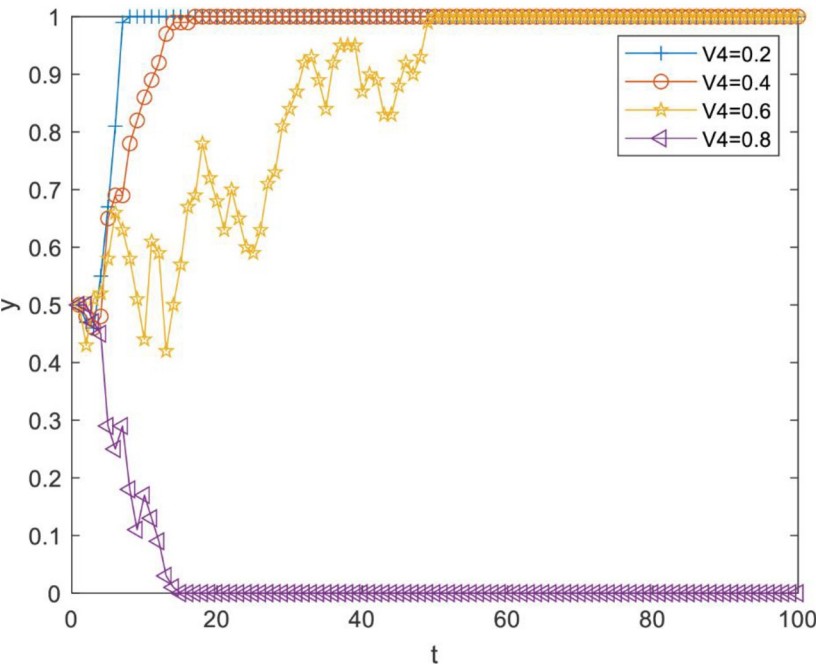

**Fig 2. Impact of the perceived cost of knowledge transfer on evolutionary results.**

than the critical value, $y$ converges to 0. When $V_i^6$ is greater than the critical value, $y$ converges to 1, and an increase in $V_i^6$ can make $y$ converge to 1 faster. Therefore, improving perceived organizational punishment value can encourage more R&D personnel to actively participate in knowledge transfer activities.

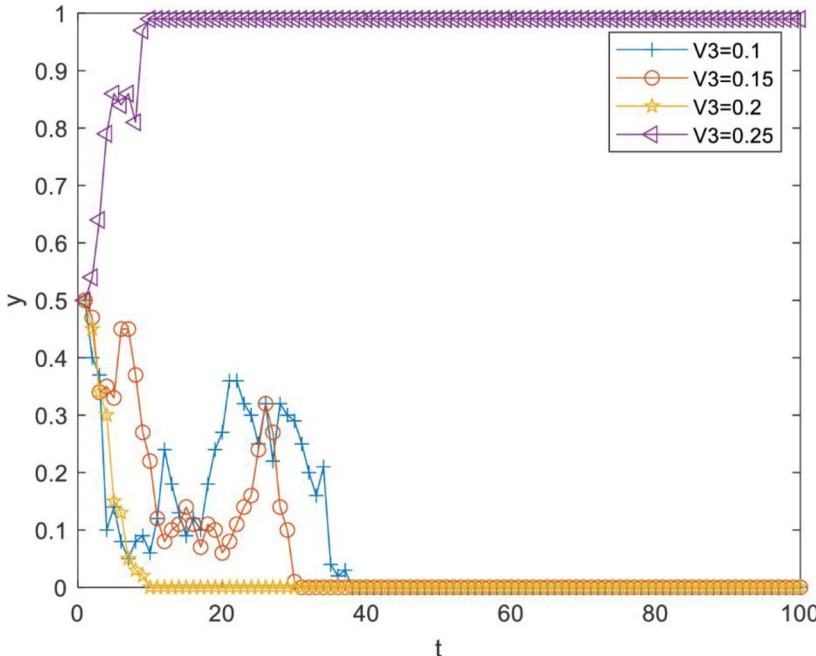

**Fig 3. Impact of perceived knowledge synergy benefit on evolutionary results.**

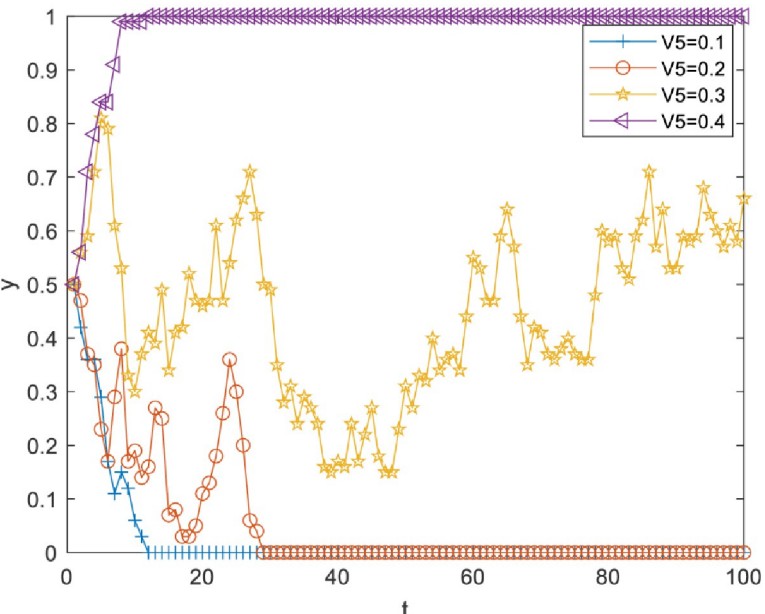

**Fig 4. Impact of perceived organizational reward value on evolutionary results.**

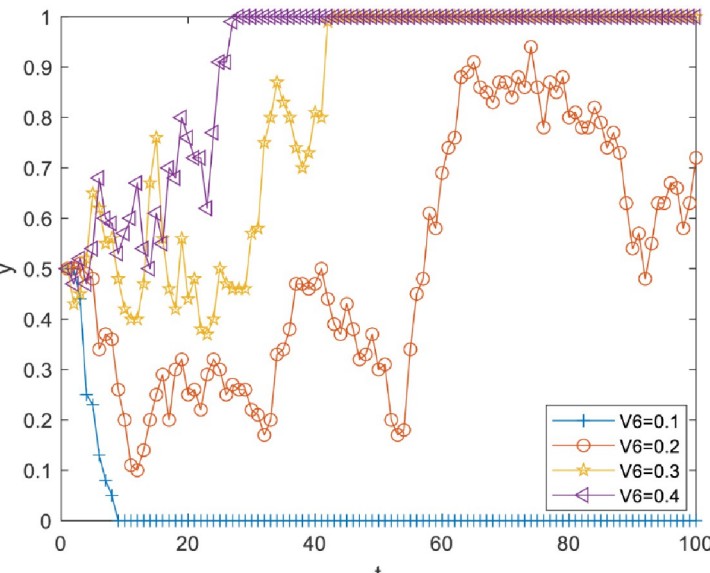

**Fig 5. Impact of perceived organizational punishment on evolutionary results.**

## Discussion

Previous studies on knowledge transfer mostly used empirical methods to examine influencing factors, but were unable to deeply explore their influencing mechanisms; or used theoretical methods to investigate the influencing mechanisms of critical factors. However, they did not always run through the assumption of bounded rationality, thus making it difficult for the

research factors to get close to the objective reality. Accordingly, this study combines the two issues and adopts the prospect value function of prospect theory to replace the benefit function of the evolutionary game benefits matrix to construct the knowledge transfer perceived benefits matrix, and based on which, it analyzes the evolutionary game mechanism of knowledge transfer actions within the R&D teams. It deduces two critical conditions and four key factors to achieve a high degree of knowledge circulation within R&D teams, analyzes the reasons behind the difficulty to achieve it in reality, and simulates the paths and modes of the influence of the key factors on the knowledge transfer evolution system.

Consequently, the research discussions are as follows.

First, the improved evolutionary game tools. Prospect theory mainly describes the cognitive behavioral characteristics of bounded rational individuals. Evolutionary game theory mainly examines the evolutionary paths and results of the decision-making behaviors of bounded rational groups. Thus, this study employs the prospect value function in prospect theory to replace the benefits function in evolutionary game theory to realize an organic combination of the two theories, constructs a perceived benefits matrix that is different from the traditional benefits matrix and applies it to evolutionary game analysis. It can better reflect the characteristics of bounded rationality in the entire process from perception to decision-making, and explain game phenomena and predict game results more reasonably. The improved evolutionary game tool has significant applicability to bounded rational games.

Second, two conditions and four key factors eliminate knowledge non-transfer behavior. Through evolutionary reasoning based on the perceived benefits matrix of knowledge transfer behaviors, we conclude that when

$$v(d_1 k_1^m k_2^n) + v(ek_1) + \pi(\eta)v(\varphi) > v(c_1 k_1),$$
$$v(d_2 k_1^m k_2^n) + v(ek_2) + \pi(\eta)v(\varphi) > v(c_2 k_2)$$

these two conditions are met simultaneously, all members of the R&D team will attach importance to knowledge transfer activities, and the system will evolve to the ideal state. Moreover, the four parameters (i.e., perceived cost, perceived synergy benefit, perceived organizational reward value, and perceived organizational punishment value) are key factors in system evolution. Third, reasons why these two conditions are difficult to achieve in reality. By combining prospect theory and behavioral research results, the analysis results show that decision-makers are usually affected by internal psychological factors, such as over confidence, reflection effect, loss avoidance effect, and obsession with a small probability event effect, which make both participants in the game overestimate costs, underestimate benefits, and be willing to take risks when facing uncertain losses. This situation makes the two conditions more difficult to achieve in practice, and the optimal evolutionary state is difficult to reach. It better explains the contradictory phenomenon in reality in which the loss of the knowledge block is significant but intentional knowledge non-transfer behaviors frequently occur.

(1) Fourth, the paths and modes of these four key factors affect the evolution of knowledge transfer behaviors. The lower the perceived cost, the higher the perceived synergy benefit, perceived organizational reward value, and perceived organizational punishment value, and the higher the proportion of members who choose knowledge transfer strategies in the R&D team. All four key factors had clear critical values. When the system parameters reach these critical values, the knowledge transfer system evolves to an ideal state.

## Conclusion and implication

### Conclusion

This study extends knowledge management in Chinese enterprises by integrating prospect theory and evolutionary game theory. We construct a knowledge transfer perceived benefits matrix, discover the critical conditions and key factors for the knowledge transfer system to achieve an ideal equilibrium, and propose the crucial role of R&D personnel's psychological awareness in a game of knowledge transfer behavior.

### Implications

This study enriches the research content and perspective on knowledge transfer and expands the theory of knowledge transfer. It helps manufacturing enterprises accurately understand the nature and law of knowledge transfer within R&D teams, including the boundary conditions, evolution process and internal influence mechanism of knowledge transfer within teams, and provides effective strategies for manufacturing enterprises to actively implement knowledge transfer to improve the innovation performance of R&D teams in manufacturing enterprises.

The findings of this study have considerable practical implications.

First, the cost of knowledge transfer and its perceived value should be reduced. The present study points to the need to improve the knowledge transfer abilities (e.g., knowledge coding ability, imparting ability, and learning ability) of R&D personnel through training, contests and other activities; standardize and humanize the organization's reward and punishment system and knowledge transfer process; and make R&D personnel aware, understand, and be familiar with knowledge transfer methods through training and publicity.

Second, knowledge synergy benefit and perceived value must be improved. Introducing R&D talents with similar backgrounds and differentiated knowledge to maintain the heterogeneity of team knowledge and optimize the knowledge structure of the R&D team; strengthening the training of R&D personnel's professional knowledge and social and team collaboration skills; creating a good atmosphere and platform for team knowledge communication and collaboration are essential.

Finally, this study's findings highlight the need to perfect the organizational knowledge transfer reward and punishment system and its deterrence; respect and support the intellectual property rights of R&D personnel, incorporate R&D personnel's performance in knowledge transfer activities into the performance evaluation index system; and establish a complete knowledge transfer incentive system. In addition, penalize behaviors that are not conducive to knowledge circulation and innovation from multiple aspects (e.g., finances, reputations, and qualifications); and reward positive behaviors that make outstanding contributions to the team's knowledge transfer activities from multiple aspects (e.g., job promotion, financial rewards, honors, and potential opportunities). Furthermore, the supervision and review of knowledge transfer work to eliminate the fluke of escaping organizational punishments should be strengthened.

### Limitations and future research

This study incorporates explicit knowledge and implicit knowledge, and the concept is broad but not sufficiently specific. Given the rapid development of the digital economy, digital technology plays a vital role in responding to public crisis events and scientific and technological innovation [60]. In future research, we will systematically explore the processes of transfer, sharing, and creation, and the influence mechanism of digital technology, which is a manifestation of knowledge application.

## Author Contributions

**Conceptualization:** Xiaoya Zhu.

**Data curation:** Xiaohua Meng.

**Formal analysis:** Xiaoya Zhu, Xiaohua Meng.

**Funding acquisition:** Xiaoya Zhu.

**Investigation:** Xiaoya Zhu, Yanjing Zhang.

**Methodology:** Xiaohua Meng, Yanjing Zhang.

**Software:** Yanjing Zhang.

**Supervision:** Xiaohua Meng.

**Visualization:** Xiaoya Zhu, Xiaohua Meng.

**Writing – original draft:** Xiaoya Zhu.

**Writing – review & editing:** Xiaoya Zhu, Xiaohua Meng.

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
