## [Decision Letter · Decision Letter 0]

7 Jun 2023

PONE-D-23-14744Evolutionary Game Study of Knowledge Transfer within R&D Team Based on Prospect TheoryPLOS ONE

Dear Dr. Meng,

Thank you for submitting your manuscript to PLOS ONE. After careful consideration, we feel that it has merit but does not fully meet PLOS ONE’s publication criteria as it currently stands. Therefore, we invite you to submit a revised version of the manuscript that addresses the points raised during the review process.

We look forward to receiving your revised manuscript.

Kind regards,

Syed Abdul Rehman Khan, PhD

Academic Editor

PLOS ONE

Journal Requirements:

   "This study was supported by the Jiangsu Province Social Science Foundation for Youths (No.21GLC012) and the National Natural Science Foundation of China (No.72174136).

Conceptualization: Xiaoya Zhu.

Data curation: Xiaohua Meng.

Formal analysis: Xiaoya Zhu, Xiaohua Meng.

Preparation of the manuscript: Xiaoya Zhu, Xiaohua Meng."

5. Please amend your authorship list in your manuscript file to include author Yanjing Zhang.

Reviewers' comments:

Reviewer's Responses to Questions

**Comments to the Author**

1. Is the manuscript technically sound, and do the data support the conclusions?

Reviewer #1: Yes

Reviewer #2: Partly

2. Has the statistical analysis been performed appropriately and rigorously? 

Reviewer #1: Yes

Reviewer #2: Yes

3. Have the authors made all data underlying the findings in their manuscript fully available?

Reviewer #1: Yes

Reviewer #2: Yes

4. Is the manuscript presented in an intelligible fashion and written in standard English?

Reviewer #1: Yes

Reviewer #2: Yes

5. Review Comments to the Author

Reviewer #1: The reviewer believes that the topic “Evolutionary Game Study of Knowledge Transfer within R&D Team Based on Prospect Theory” is worthy of investigation. However, the following needs to be addressed. There are minor and major issues that should be corrected. I believe the paper could be further strengthened by added information about.

1.The title does not provide a core theme of the topic.

2.Please add numerical simulation part and specify the source of the simulation data.

3.The language of this manuscript needs help from native speakers.

4.Please underscore the scientific value-added to your paper in your abstract. Your abstract should clearly state the essence of the problem you are addressing, what you did and what you found and recommend. That would help a prospective reader of the abstract to decide if they wish to read the entire article.

5.For enterprises, to achieve collaborative innovation............ This a very vague statement. These sentences do not provide any information on how the concept could be conceptualized? - The Introduction should have 1) a concise but complete justification of the topic's importance both academically and practically, and 2) an explanation of the gaps both in research and practice. Please review appropriate literature in the Introduction, with the research question clearly arising from that review.

6.What authors wanted to convey. Here author must build research gap following the previous studies.-The manuscript does not answer the following concerns: Why is it timeliness to explore such a study? What makes this study different from the previously published studies? Are there any similarly findings in line with the previously published studies? Are the findings different from prior academic studies that were conducted elsewhere, if any? What it requires, what are the new technologies, some recent issue highlights the importance. See the following: An adoption-implementation framework of digital green knowledge to improve the performance of digital green innovation practices for industry 5.0.

7.-There is no flow in the text. It partly depends on the lack of proofreading but also on the fact that many statements and claims are made without being followed up by a clear and logical discussion. It is especially problematic in the Introduction that brings up a number of findings from different areas without linking them together.

8.-More importantly, the choice of the questionnaire questions should be explained in light of the theory and the prior literature on the topic. The arguments are simply relationships and causes very close to the replication of many studies dealing with the same thing. For example, what is connection of upgrading path of manufacturing enterprises from the value perspective. See the following: How to Improve the Quality and Speed of Green New Product Development? Processes 2019, 7, 443. https://doi.org/10.3390/pr7070443

9.-Methodology: Model.. I suggest authors here build your main heading on Research and data methodology. Clearly explain the model building process, and what previous studies have used similar models (model testing approach).

See the following: A three-player game model for promoting the diffusion of green technology in manufacturing enterprises from the perspective of supply and demand.https://doi.org/10.3390/math8091585

A stochastic differential game of low carbon technology sharing in collaborative innovation system of superior enterprises and inferior enterprises under uncertain environment, https://doi.org/10.1515/math-2018-0056

10.The authors should emphasize the important role of digital technology in future research. See the following: Information fusion for future COVID-19 prevention: continuous mechanism of big data intelligent innovation for the emergency management of a public epidemic outbreak.

11.Please consider this structure for manuscript final part.

-Discussion

-Conclusion

-Managerial Implication

-Practical/Social Implications

12.Please make sure your conclusions' section underscores the scientific value-added of your paper, and/or the applicability of your findings/results. Highlight the novelty of your study. In addition to summarizing the actions taken and results, please strengthen the explanation of their significance. It is recommended to use quantitative reasoning comparing with appropriate benchmarks, especially those stemming from previous work.

Reviewer #2: Comments:

• The manuscript exhibits technical soundness, and the data support the conclusions drawn. The manuscript presents scientifically rigorous research backed by empirical data substantiating the findings. Rigorous experimentation has been conducted, utilizing appropriate controls, replication techniques, and sample size. But the conclusion section is not satisfactory that’s why it is advised to the authors that they should extend the conclusion.

• The statistical analysis has been executed with appropriate and rigorous methods. However, it is recommended that authors include equation numbers for clarification.

• The data must be included as an integral component of the manuscript or its supplementary information or, alternatively, deposited in a publicly accessible repository. Kindly include the "Data availability statement" in your manuscript.

• The manuscript is written in standard English and is presented understandably.

• It is recommended that authors include recent studies about the topic in concern. Therefore, it is advised to incorporate the following references to enhance the existing literature:

• https://doi.org/10.1007/s10661-022-10835-w

• https://doi.org/10.1016/j.egyr.2021.11.231

• https://doi.org/10.1016/j.jenvman.2023.117968

• https://doi.org/10.1177/0734242X221149329

• https://doi.org/10.1016/j.jenvman.2022.114570

6. PLOS authors have the option to publish the peer review history of their article (what does this mean?). If published, this will include your full peer review and any attached files.

Reviewer #1: No

Reviewer #2: **Yes: **Muhammad Tabish

---

## [Author Response · Author response to Decision Letter 0]

1 Jul 2023

Editor: 

1. Please ensure that your manuscript meets PLOS ONE's style requirements, including those for file naming. The PLOS ONE style templates can be found at https://journals.plos.org/plosone/s/file?id=wjVg/PLOSOne_formatting_sample_main_body.pdf and https://journals.plos.org/plosone/s/file?id=ba62/PLOSOne formatting _sample_title_authors_affiliations.pdf

Response: The manuscript has been carefully proofread and corrected to meet PLOS ONE’s style requirements.

Response: The funders had no role in study design, data collection and analysis, decision to publish, or preparation of the manuscript.

Response: All relevant data are within the paper.

Response: ORCID ID for the corresponding author: 0000-0002-4938-3458.

5. Please amend your authorship list in your manuscript file to include author Yanjing Zhang.

Response: We have added co-author Yanjing Zhang in the authorship list in our manuscript.

Reviewer #1:

1.The title does not provide a core theme of the topic.

Response: To highlight the core theme, the title was changed to “How to Promote Knowledge Transfer within R&D Team? An Evolutionary Game based on Prospect Theory.”

2.Please add numerical simulation part and specify the source of the simulation data.

Response: Many thanks for your advice. We have extended the numerical simulation part and added more details about the simulation data as follows:

Numerical simulation

According to the analysis results of the evolutionary game, within R&D teams, the perceived cost, perceived synergy benefit, perceived organizational reward value, and perceived organizational punishment determine the final evolution result of the knowledge transfer system to a certain extent. This study uses MATLAB to simulate the evolution process of knowledge transfer within R&D teams and reveal the influence paths and modes of the four key factors on knowledge transfer. 

The fundamental purpose of knowledge transfer is to realize the full sharing and circulation of knowledge in R&D teams. Hence, this study uses the proportion y ([0,1]) of the number of R&D personnel who adopt knowledge transfer strategies to the total number of R&D personnel in the R&D team as the metrics. In addition, it tests each set of parameters 50 times to obtain stable simulation results.

Based on the actual situation and the basic assumptions of the model, the initial values of simulation parameters were set as .

(1) Impact of perceived cost of knowledge transfer on evolutionary results. was set as 0.2, 0.4, 0.6 and 0.8. As Fig 2 illustrates, there is a critical value between 0.6 and 0.8. When is less than the critical value, converges to 1, but the increase of can slow down the speed of converging to 1; when is more than the critical value, converges to 0. Therefore, reducing the perceived cost of knowledge transfer between R&D personnel can effectively promote knowledge transfer.

Fig 2. Impact of the perceived cost of knowledge transfer on evolutionary results

(1) Impact of perceived knowledge synergy benefit on the evolutionary results. was set as 0.1, 0.15, 0.2 and 0.25. As Fig 3 illustrates, there is a critical value between 0.2 and 0.25. When is less than the critical value, y converges to 0, and when is more than the critical value, y converges to 1. Therefore, R&D personnel can be effectively motivated to choose knowledge transfer strategies only when the perceived knowledge synergy benefit increases above a critical value. 

Fig 3. Impact of perceived knowledge synergy benefit on evolutionary results

(2) Impact of the perceived organizational reward value on evolutionary results. was set as 0.1, 0.2, 0.3 and 0.4. As Fig 4 illustrates, there is a critical value between 0.2 and 0.3. When is less than the critical value, y converges to 0, and the increase of reduces the speed at which y converges to 0. When is more than the critical value, y converges to 1, and the increase of can make y converge to 1 faster. Therefore, improving the perceived organizational reward value of R&D team members can prompt more R&D personnel to pay attention to knowledge transfer activities.

Fig 4. Impact of perceived organizational reward value on evolutionary results

(3) Impact of perceived organizational punishment value on evolutionary results. was set as 0.1, 0.2, 0.3 and 0.4. Fig 5 reveals a critical value between 0.1 and 0.2. When is less than the critical value, y converges to 0. When is greater than the critical value, y converges to 1, and an increase in can make y converge to 1 faster. Therefore, improving perceived organizational punishment value can encourage more R&D personnel to actively participate in knowledge transfer activities.

Fig 5. Impact of perceived organizational punishment on evolutionary results

3.The language of this manuscript needs help from native speakers.

Response: We have asked language help from the native speakers. The English language in the revised manuscript has been carefully corrected to improve grammar and readability. 

4.Please underscore the scientific value-added to your paper in your abstract. Your abstract should clearly state the essence of the problem you are addressing, what you did and what you found and recommend. That would help a prospective reader of the abstract to decide if they wish to read the entire article.

Response: Thank you very much for your review and comments. According to your advice, we revised the abstract section as follows: Knowledge transfer is the basis for R&D teams and enterprises to improve innovation performance, win market competition and seek sustainable development. In order to explore the path to promote knowledge transfer within the R&D team, this study considers the bounded rationality and risk preference of individuals, incorporates prospect theory into evolutionary game, constructs a perceived benefits matrix distinct from the traditional benefits matrix, and simulates the evolutionary game process. The results show that, R&D personnel’s knowledge transfer decisions depend on the net income difference among strategies; only if perceived cost is less than the sum of perceived synergy benefit, perceived organization reward value, and perceived organization punishment value, can knowledge be fully shared and transferred within the R&D team. Moreover, R&D personnel’s knowledge transfer decisions are interfered by the irrational psychological factors, including overconfidence, reflection, loss avoidance, and obsession with small probability events. The findings help R&D teams achieve breakthroughs in improving the efficiency of knowledge transfer, thereby enhancing the capacity of enterprises for collaborative innovation.

5.For enterprises, to achieve collaborative innovation............ This a very vague statement. These sentences do not provide any information on how the concept could be conceptualized? - The Introduction should have 1) a concise but complete justification of the topic's importance both academically and practically, and 2) an explanation of the gaps both in research and practice. Please review appropriate literature in the Introduction, with the research question clearly arising from that review.

Response: Thank you for your comments. According to your advice, we added some additional information as follows: Knowledge transfer is an important means to achieve collaborative innovation [1]. Especially, knowledge transfer within R&D team, as an important organization for enterprises to carry out scientific research and knowledge innovation, is the key way to realize knowledge collaboration and resource complementarity among team members [2-5]. Whether knowledge transfer can be carried out continuously and stably determines the team’s knowledge innovation ability and the enterprise's scientific research and technological innovation potential, which is of great significance to the rapid development of enterprises and the formation of innovation system [6]. However, in practice, knowledge transfer activities in Chinese enterprises R&D teams are inefficient, R&D personnel’s willingness to transfer knowledge and absorptive capacity are insufficient, thus constraining the knowledge transfer process and team collaborative innovation [7]. Accordingly, this situation poses the following questions. how to promote knowledge transfer within R&D teams? What mechanism underlies knowledge transfer? What are the key factors influencing R&D personnel’s knowledge transfer decisions?

6.What authors wanted to convey. Here author must build research gap following the previous studies. -The manuscript does not answer the following concerns: Why is it timeliness to explore such a study? What makes this study different from the previously published studies? Are there any similarly findings in line with the previously published studies? Are the findings different from prior academic studies that were conducted elsewhere, if any? What it requires, what are the new technologies, some recent issue highlights the importance. See the following: An adoption-implementation framework of digital green knowledge to improve the performance of digital green innovation practices for industry 5.0.

Response: Thank you for your comments. We have read and studied the article carefully, and added some additional information as follows: In addition, Yin (2019, 2022) and Khan (2023) noted that the knowledge spiral has a significant positive effect on new green product development speed and quality, and digital green knowledge creation significantly affects digital green innovation performance [29-31]. This finding provides a perfect theory to understand knowledge transfer better. However, through numerous literature reviews, it is clear that few scholars have studied the deep process mechanism of knowledge transfer within the R&D team the exact conditions for R&D personnel to choose knowledge transfer behaviors. It is a key research gap. According to knowledge management theory, knowledge transfer is a game process and the relationship between subjects is a kind of coopetition [32, 33]. The transfer subjects expect to obtain the return of knowledge transfer while worrying that the existing competitive advantage would be weakened [32-35]. This fact implies that, from the perspective of individuals, knowledge transfer is a risky behavior and can produce a win-win situation while also bring risks [34-37]. As game theory effectively analyzes the decision-making process [38-41], some scholars have begun adopting game theory to explore knowledge transfer behavior [32, 33, 42-45]. However, the traditional evolutionary game theory cannot fully explain a subject’s irrationality and risk preference.

……

The contributions of this paper are as follows. From a micro perspective, it reveals the process mechanism of knowledge transfer within an R&D team by integrating prospect theory and evolutionary game theory, clarifies the difference between the perceived and theoretical benefits of the knowledge transfer behavior of R&D personnel, discovers the critical conditions and key factors for the knowledge transfer system to achieve an ideal equilibrium, and proposes the crucial role of R&D personnel’s psychological awareness in the knowledge transfer game. Ultimately, this study can help enterprises understand why promoting knowledge transfer in practice is difficult and what measures can effectively motivate R&D personnel to make knowledge transfer decisions.

7.-There is no flow in the text. It partly depends on the lack of proofreading but also on the fact that many statements and claims are made without being followed up by a clear and logical discussion. It is especially problematic in the Introduction that brings up a number of findings from different areas without linking them together.

Response: Many thanks for your advice. According to your advice, we re-sorted out the logic and expression of the full text to make it more logical and coherent, especially in the Introduction, we rearranged the logical relationships of the existing literature and reviewed them again.

8.-More importantly, the choice of the questionnaire questions should be explained in light of the theory and the prior literature on the topic. The arguments are simply relationships and causes very close to the replication of many studies dealing with the same thing. For example, what is connection of upgrading path of manufacturing enterprises from the value perspective. See the following: How to Improve the Quality and Speed of Green New Product Development? Processes 2019, 7, 443. https://doi.org/10.3390/pr7070443

Response: Many thanks for your advice and the literature. We have read and studied the article carefully, and added more details in the Introduction as follows: In addition, Yin (2019, 2022) and Khan (2023) noted that the knowledge spiral has a significant positive effect on new green product development speed and quality, and digital green knowledge creation significantly affects digital green innovation performance [29-31]. This finding provides a perfect theory to understand knowledge transfer better. However, through numerous literature reviews, it is clear that few scholars have studied the deep process mechanism of knowledge transfer within the R&D team the exact conditions for R&D personnel to choose knowledge transfer behaviors. It is a key research gap. According to knowledge management theory, knowledge transfer is a game process and the relationship between subjects is a kind of coopetition [32, 33]. The transfer subjects expect to obtain the return of knowledge transfer while worrying that the existing competitive advantage would be weakened [32-35]. This fact implies that, from the perspective of individuals, knowledge transfer is a risky behavior and can produce a win-win situation while also bring risks [34-37]. As game theory effectively analyzes the decision-making process [38-41], some scholars have begun adopting game theory to explore knowledge transfer behavior [32, 33, 42-45]. However, the traditional evolutionary game theory cannot fully explain a subject’s irrationality and risk preference.

9.-Methodology: Model.. I suggest authors here build your main heading on Research and data methodology. Clearly explain the model building process, and what previous studies have used similar models (model testing approach).

See the following: A three-player game model for promoting the diffusion of green technology in manufacturing enterprises from the perspective of supply and demand.https://doi.org/10.3390/math8091585

A stochastic differential game of low carbon technology sharing in collaborative innovation system of superior enterprises and inferior enterprises under uncertain environment, https://doi.org/10.1515/math-2018-0056

Response: Many thanks for your comments. We have read and studied the articles carefully, and amended the main headings based on the model building processes: 

An evolutionary game analysis of knowledge transfer Evolutionary game model

Game model construction

(1) Game player

……

(2) Behavioral strategies of players

……

(3) Model hypothesis

Hypothesis 1: ……

Hypothesis 2: ……

Hypothesis 3: ……

Hypothesis 4: ……

Hypothesis 5: ……

Hypothesis 6: ……

……

10.The authors should emphasize the important role of digital technology in future research. See the following: Information fusion for future COVID-19 prevention: continuous mechanism of big data intelligent innovation for the emergency management of a public epidemic outbreak.

Response: Thank you for your comments. According to your advice, we have emphasized the important role of digital technology in future research at the end of the study: This study incorporates explicit knowledge and implicit knowledge, and the concept is broad but not sufficiently specific. Given the rapid development of the digital economy, digital technology plays a vital role in responding to public crisis events and scientific and technological innovation [60]. In future research, we will systematically explore the processes of transfer, sharing, and creation, and the influence mechanism of digital technology, which is a manifestation of knowledge application.

11.Please consider this structure for manuscript final part.

-Discussion

-Conclusion

-Managerial Implication

-Practical/Social Implications

Response: Thank you for your advice. We have revised the structure for manuscript final part: 

-Discussion 

-Conclusion and implication

Conclusion

Implications

-Limitations and future research

12.Please make sure your conclusions' section underscores the scientific value-added of your paper, and/or the applicability of your findings/results. Highlight the novelty of your study. In addition to summarizing the actions taken and results, please strengthen the explanation of their significance. It is recommended to use quantitative reasoning comparing with appropriate benchmarks, especially those stemming from previous work.

Response: Thank you for your comments. According to your advice, more content has been added as follows: 

Discussion 

Previous studies on knowledge transfer mostly used empirical methods to examine influencing factors, but were unable to deeply explore their influencing mechanisms; or used theoretical methods to investigate the influencing mechanisms of critical factors. However, they did not always run through the assumption of bounded rationality, thus making it difficult for the research factors to get close to the objective reality. Accordingly, this study combines the two issues and adopts the prospect value function of prospect theory to replace the benefit function of the evolutionary game benefits matrix to construct the knowledge transfer perceived benefits matrix, and based on which, it analyzes the evolutionary game mechanism of knowledge transfer actions within the R&D teams. It deduces two critical conditions and four key factors to achieve a high degree of knowledge circulation within R&D teams, analyzes the reasons behind the difficulty to achieve it in reality, and simulates the paths and modes of the influence of the key factors on the knowledge transfer evolution system. 

Consequently, the research discussions are as follows.

First, the improved evolutionary game tools. Prospect theory mainly describes the cognitive behavioral characteristics of bounded rational individuals. Evolutionary game theory mainly examines the evolutionary paths and results of the decision-making behaviors of bounded rational groups. Thus, this study employs the prospect value function in prospect theory to replace the benefits function in evolutionary game theory to realize an organic combination of the two theories, constructs a perceived benefits matrix that is different from the traditional benefits matrix and applies it to evolutionary game analysis. It can better reflect the characteristics of bounded rationality in the entire process from perception to decision-making, and explain game phenomena and predict game results more reasonably. The improved evolutionary game tool has significant applicability to bounded rational games.

Second, two conditions and four key factors eliminate knowledge non-transfer behavior. Through evolutionary reasoning based on the perceived benefits matrix of knowledge transfer behaviors, we conclude that when

these two conditions are met simultaneously, all members of the R&D team will attach importance to knowledge transfer activities, and the system will evolve to the ideal state. Moreover, the four parameters (i.e., perceived cost, perceived synergy benefit, perceived organizational reward value, and perceived organizational punishment value) are key factors in system evolution. Third, reasons why these two conditions are difficult to achieve in reality. By combining prospect theory and behavioral research results, the analysis results show that decision-makers are usually affected by internal psychological factors, such as over confidence, reflection effect, loss avoidance effect, and obsession with a small probability event effect, which make both participants in the game overestimate costs, underestimate benefits, and be willing to take risks when facing uncertain losses. This situation makes the two conditions more difficult to achieve in practice, and the optimal evolutionary state is difficult to reach. It better explains the contradictory phenomenon in reality in which the loss of the knowledge block is significant but intentional knowledge non-transfer behaviors frequently occur.

(1) Fourth, the paths and modes of these four key factors affect the evolution of knowledge transfer behaviors. The lower the perceived cost, the higher the perceived synergy benefit, perceived organizational reward value, and perceived organizational punishment value, and the higher the proportion of members who choose knowledge transfer strategies in the R&D team. All four key factors had clear critical values. When the system parameters reach these critical values, the knowledge transfer system evolves to an ideal state.

Conclusion and implication

Conclusion

This study extends knowledge management in Chinese enterprises by integrating prospect theory and evolutionary game theory. We construct a knowledge transfer perceived benefits matrix, discover the critical conditions and key factors for the knowledge transfer system to achieve an ideal equilibrium, and propose the crucial role of R&D personnel’s psychological awareness in a game of knowledge transfer behavior.

Implications

This study enriches the research content and perspective on knowledge transfer and expands the theory of knowledge transfer. It helps manufacturing enterprises accurately understand the nature and law of knowledge transfer within R&D teams, including the boundary conditions, evolution process and internal influence mechanism of knowledge transfer within teams, and provides effective strategies for manufacturing enterprises to actively implement knowledge transfer to improve the innovation performance of R&D teams in manufacturing enterprises. 

The findings of this study have considerable practical implications.

First, the cost of knowledge transfer and its perceived value should be reduced. The present study points to the need to improve the knowledge transfer abilities (e.g., knowledge coding ability, imparting ability, and learning ability) of R&D personnel through training, contests and other activities; standardize and humanize the organization’s reward and punishment system and knowledge transfer process; and make R&D personnel aware, understand, and be familiar with knowledge transfer methods through training and publicity. 

Second, knowledge synergy benefit and perceived value must be improved. Introducing R&D talents with similar backgrounds and differentiated knowledge to maintain the heterogeneity of team knowledge and optimize the knowledge structure of the R&D team; strengthening the training of R&D personnel's professional knowledge and social and team collaboration skills; creating a good atmosphere and platform for team knowledge communication and collaboration are essential. 

Finally, this study’s findings highlight the need to perfect the organizational knowledge transfer reward and punishment system and its deterrence; respect and support the intellectual property rights of R&D personnel, incorporate R&D personnel’s performance in knowledge transfer activities into the performance evaluation index system; and establish a complete knowledge transfer incentive system. In addition, penalize behaviors that are not conducive to knowledge circulation and innovation from multiple aspects (e.g., finances, reputations, and qualifications); and reward positive behaviors that make outstanding contributions to the team’s knowledge transfer activities from multiple aspects (e.g., job promotion, financial rewards, honors, and potential opportunities). Furthermore, the supervision and review of knowledge transfer work to eliminate the fluke of escaping organizational punishments should be strengthened.

Reviewer #2:

1. The manuscript exhibits technical soundness, and the data support the conclusions drawn. The manuscript presents scientifically rigorous research backed by empirical data substantiating the findings. Rigorous experimentation has been conducted, utilizing appropriate controls, replication techniques, and sample size. But the conclusion section is not satisfactory that’s why it is advised to the authors that they should extend the conclusion.

Response: Many thanks for your comments. According to your advice, we have extended the conclusion as follows: 

Conclusion and implication

Conclusion

This study extends knowledge management in Chinese enterprises by integrating prospect theory and evolutionary game theory. We construct a knowledge transfer perceived benefits matrix, discover the critical conditions and key factors for the knowledge transfer system to achieve an ideal equilibrium, and propose the crucial role of R&D personnel’s psychological awareness in a game of knowledge transfer behavior.

Implications

This study enriches the research content and perspective on knowledge transfer and expands the theory of knowledge transfer. It helps manufacturing enterprises accurately understand the nature and law of knowledge transfer within R&D teams, including the boundary conditions, evolution process and internal influence mechanism of knowledge transfer within teams, and provides effective strategies for manufacturing enterprises to actively implement knowledge transfer to improve the innovation performance of R&D teams in manufacturing enterprises. 

The findings of this study have considerable practical implications.

First, the cost of knowledge transfer and its perceived value should be reduced. The present study points to the need to improve the knowledge transfer abilities (e.g., knowledge coding ability, imparting ability, and learning ability) of R&D personnel through training, contests and other activities; standardize and humanize the organization’s reward and punishment system and knowledge transfer process; and make R&D personnel aware, understand, and be familiar with knowledge transfer methods through training and publicity. 

Second, knowledge synergy benefit and perceived value must be improved. Introducing R&D talents with similar backgrounds and differentiated knowledge to maintain the heterogeneity of team knowledge and optimize the knowledge structure of the R&D team; strengthening the training of R&D personnel's professional knowledge and social and team collaboration skills; creating a good atmosphere and platform for team knowledge communication and collaboration are essential. 

Finally, this study’s findings highlight the need to perfect the organizational knowledge transfer reward and punishment system and its deterrence; respect and support the intellectual property rights of R&D personnel, incorporate R&D personnel’s performance in knowledge transfer activities into the performance evaluation index system; and establish a complete knowledge transfer incentive system. In addition, penalize behaviors that are not conducive to knowledge circulation and innovation from multiple aspects (e.g., finances, reputations, and qualifications); and reward positive behaviors that make outstanding contributions to the team’s knowledge transfer activities from multiple aspects (e.g., job promotion, financial rewards, honors, and potential opportunities). Furthermore, the supervision and review of knowledge transfer work to eliminate the fluke of escaping organizational punishments should be strengthened.

2. The statistical analysis has been executed with appropriate and rigorous methods. However, it is recommended that authors include equation numbers for clarification.

Response: Thank you for your comments. According to your advice, we added numbers to all the equations.

3. The data must be included as an integral component of the manuscript or its supplementary information or, alternatively, deposited in a publicly accessible repository. Kindly include the "Data availability statement" in your manuscript.

Response: Thank you for your comments. According to your advice, we add a section “Data availability statement” as follows:

Data availability statement

All relevant data are within the paper.

4. The manuscript is written in standard English and is presented understandably.

Many thanks for your comments. You give us much confidence.

5. It is recommended that authors include recent studies about the topic in concern. Therefore, it is advised to incorporate the following references to enhance the existing literature:

• https://doi.org/10.1007/s10661-022-10835-w

• https://doi.org/10.1016/j.egyr.2021.11.231

• https://doi.org/10.1016/j.jenvman.2023.117968

• https://doi.org/10.1177/0734242X221149329

• https://doi.org/10.1016/j.jenvman.2022.114570

Response: Many thanks for the five valuable articles, and we have studied and read them carefully. According to your advice, we have incorporated the above references to enhance the existing literature.

---

## [Decision Letter · Decision Letter 1]

18 Jul 2023

How to Promote Knowledge Transfer within R&D Team? An Evolutionary Game based on Prospect Theory

PONE-D-23-14744R1

Dear Dr. Meng,

We’re pleased to inform you that your manuscript has been judged scientifically suitable for publication and will be formally accepted for publication once it meets all outstanding technical requirements.

Kind regards,

Syed Abdul Rehman Khan, PhD

Academic Editor

PLOS ONE

Reviewers' comments:

Reviewer's Responses to Questions

**Comments to the Author**

1. If the authors have adequately addressed your comments raised in a previous round of review and you feel that this manuscript is now acceptable for publication, you may indicate that here to bypass the “Comments to the Author” section, enter your conflict of interest statement in the “Confidential to Editor” section, and submit your "Accept" recommendation.

Reviewer #1: (No Response)

Reviewer #2: All comments have been addressed

2. Is the manuscript technically sound, and do the data support the conclusions?

Reviewer #1: (No Response)

Reviewer #2: Yes

3. Has the statistical analysis been performed appropriately and rigorously? 

Reviewer #1: (No Response)

Reviewer #2: Yes

4. Have the authors made all data underlying the findings in their manuscript fully available?

Reviewer #1: (No Response)

Reviewer #2: Yes

5. Is the manuscript presented in an intelligible fashion and written in standard English?

Reviewer #1: (No Response)

Reviewer #2: Yes

6. Review Comments to the Author

Reviewer #1: The manuscript has significantly improved as compared to the previous version. Indeed, the authors tried to improve it, and the main weaknesses are solved.

Thus, in my opinion, the manuscript is recommendable for publication.

Reviewer #2: (No Response)

7. PLOS authors have the option to publish the peer review history of their article (what does this mean?). If published, this will include your full peer review and any attached files.

Reviewer #1: No

Reviewer #2: **Yes: **Muhammad Tabish

---

## [Editor Report · Acceptance letter]

25 Jul 2023

PONE-D-23-14744R1 

How to Promote Knowledge Transfer within R&D Team? An Evolutionary Game based on Prospect Theory 

Dear Dr. Meng:

I'm pleased to inform you that your manuscript has been deemed suitable for publication in PLOS ONE. Congratulations! Your manuscript is now with our production department. 

Kind regards, 

on behalf of

Dr. Syed Abdul Rehman Khan 

Academic Editor

PLOS ONE